# Optimization of Cyclodextrin-Assisted Extraction of Phenolics from *Helichrysum italicum* for Preparation of Extracts with Anti-Elastase and Anti-Collagenase Properties

**DOI:** 10.3390/metabo13020257

**Published:** 2023-02-09

**Authors:** Marijan Marijan, Dora Tomić, Jakub W. Strawa, Lejsa Jakupović, Suzana Inić, Mario Jug, Michał Tomczyk, Marijana Zovko Končić

**Affiliations:** 1Department of Pharmacognosy, Faculty of Pharmacy and Biochemistry, University of Zagreb, A. Kovačića 1, 10000 Zagreb, Croatia; 2Department of Pharmacognosy, Faculty of Pharmacy with the Division of Laboratory Medicine, Medical University of Białystok, ul. Mickiewicza 2a, 15-230 Białystok, Poland; 3Department of Analytical Chemistry, Faculty of Pharmacy and Biochemistry, University of Zagreb, A. Kovačića 1, 10000 Zagreb, Croatia; 4Department of Pharmaceutical Technology, Faculty of Pharmacy and Biochemistry, University of Zagreb, A. Kovačića 1, 10000 Zagreb, Croatia

**Keywords:** anti-collagenase activity, anti-elastase activity, arzanol, cyclodextrins, flavonoids, *Helichrysum italicum*, LC-MS, phenolic acids, ultrasound assisted extraction

## Abstract

*Helichrysum italicum* is a plant traditionally used for skin-related disorders that is becoming an increasingly popular ingredient in cosmetic products. In this work, a “green” ultrasound-assisted extraction method for *H. italicum* phenolics was developed using skin-friendly cyclodextrins (CDs). Extraction conditions needed for the greatest yield of target compounds (total phenolics, phenolic acids, and flavonoids) were calculated. The composition of the extracts was determined using LC-MS and spectrophotometric methods. Among the tested CDs, 2-hydroxylpropyl-beta-CD (HP-β-CD) was the best suited for extraction of target phenolics and used to prepare two optimized extracts, OPT 1 (the extract with the highest phenolic acid content) and OPT 2 (the extract with the highest total phenol and flavonoid content). The extracts were prepared at 80 °C, using 0.089 g of plant material/g solvent (0.6 mmol of HP-β-CD), with or without addition of 1.95% (*w/w*) lactic acid. The main metabolite in both extracts was 3,5-*O*-dicaffeoylquinic acid. It was found that the addition of lactic acid greatly contributes to the extraction of arzanol, a well-known anti-inflammatory agent. IC_50_ values of the anti-elastase (22.360 ± 0.125 μL extract/mL and 20.067 ± 0.975 for OPT-1 and OPT-2, respectively) and anti-collagenase (12.035 ± 1.029 μL extract/mL and 14.392 ± 0.705 μL extract/mL for OPT-1 and OPT-2, respectively) activities of the extracts surpassed those of the applied positive controls, namely ursolic and gallic acids. This activity deems the prepared extracts promising ingredients for natural cosmetics, appropriate for direct use in cosmetic products, removing the need for the evaporation of conventional solvents.

## 1. Introduction

*Helichrysum italicum* (Roth) G. Don, Asteraceae (immortelle) is a Mediterranean plant and represents a rich source of versatile bioactive compounds with potential benefits for human health. It continues to play an important role not only in the traditional medicine of Mediterranean countries but also in medicinal plant-based economies worldwide. Traditionally, the plant is mostly used for respiratory, digestive and dermatological disorders, including allergies, colds, cough, liver and gallbladder disorders, sleeplessness, skin inflammation, and infections [1]. However, most recently, its use in cosmetic products has been experiencing an unprecedented rise. It is widely considered that *H. italicum*-based products can efficiently delay and even prevent skin aging [2]. Indeed, numerous studies have shown that essential oil, phenolic and other compounds, present in the plant and its extracts, are desirable preservatives and functional ingredients in cosmetic products due to their antioxidant and antibacterial activity [2] and their ability to decrease skin irritation [3] and accelerate wound healing in vivo [4].

*H. italicum* contains bioactive secondary metabolites including hydroxycinnamic acids (such as rosmarinic, neochlorogenic, isochlorogenic B, cichoric [2] 3,4-dicaffeoylquinic acid, chlorogenic acid, and 3,5-dicaffeoylquinic acid [5]), hydroxybenzoic acids (e.g., 3,4-dihydroxybenzoic, 2,4-dihydroxybenzoic, and vanillic acid) [6], flavonoids (such as rutoside [2], tiliroside [6], and kaempferol 3-*O*-glucopyranoside [5]), and coumarins (e.g., scopoletin [6]). In addition, *H. italicum* contains pyrone derivatives, with arzanol being the most prominent one [6]. Arzanol has been reported to inhibit inflammatory transcription factor nuclear factor kappa light chain enhancer of activated B cells (NF-κB) activation, HIV replication in T-cells, releases of interleukins (IL) (IL-1 β, IL-6, and IL-8), and tumor necrosis factor (TNF-α), as well as biosynthesis of Prostaglandin E2 (PGE_2_). In addition, it displays antioxidant and anti-HIV activity [7]. Besides the abundance of non-volatile metabolites, *H. italicum* is well-appreciated for its essential oil. The oil is particularly rich in neryl acetate and *α*-pinene. In addition, the oil also contains italidione I and II, *ar*- and *γ*-curcumene, *β*- and *α*-selinene, nerol, limonene, and linalool [8].

Plant extracts are becoming increasingly popular ingredients in cosmetic products. They are perceived as functional ingredients and delay or prevent processes that negatively influence skin health and appearance, for example, by protecting the skin macromolecules against enzymatic degradation induced by aging or exposure to environmental stressors [9]. However, to incorporate bioactive plant metabolites into the cosmetic formulations, they first must be efficiently extracted from plant material. One of the fast-evolving areas of medicinal plant research is the design of green and sustainable extraction methods for bioactive natural products. Besides high dissolving power, the ideal solvent should be safe, both to humans and the environment [10,11].

Among such solvents, a prominent place is occupied by aqueous solutions of cyclodextrins (CDs). CDs are water-soluble, nontoxic, and nonreducing cyclic oligosaccharides made up of d-glucopyranoside units linked by α-1,4-glycosidic bonds. The positions of hydroxyl groups provide CDs with a hydrophilic exterior, while their interior cavity is hydrophobic because of the glycoside bond orientations. Such structure enables them to form inclusion complexes with various organic compounds, including those of natural origin. Consequently, CDs may enhance the solubility and stability of small molecules, as well as reduce their unpleasant odor and agglomeration [12]. According to the number of glucose units in their structure, there are three main types of CDs: α-CD, β-CD, and γ-CD, with six, seven, and eight glucose subunits in a ring [13], respectively. Furthermore, addition of polar side-chains, such as additional hydroxypropyl groups, may yield CD molecules with improved water solubility, such as (2-hydroxypropyl)-β-CD (HP-β-CD) and (2-hydroxypropyl)-γ-CD (HP-γ-CD). Such structural features can also improve the stability of CD complexes with small molecules, as demonstrated in HP-β-CD and HP-γ-CD complexes with rutin [14]. CDs were proven to be excellent encapsulating agents for bioactive plant polyphenols such as catechin, epicatechin, morin, and quercetin [15]. Furthermore, CD-assisted extraction (CDAE) of plant phenols was proven to be a green and cost-effective alternative to traditional extraction with organic solvents, as exemplified by extraction of isoflavonoids from *Trifolium pratense* [16]. In addition to being biocompatible with dermopharmaceutical and cosmetics formulations, CDs may enhance the bioavailability [14] and passage of active molecules through the skin [17]. It has been repeatedly shown that the addition of CD to water improves the extraction efficiency for plant phenols and reduces the extraction time. In addition, the obtained phenolic extracts may retain their activity (e.g., antioxidant activity). The fact that CDs may be incorporated into the final products makes the CDAE of bioactive molecules from medicinal plants highly attractive from the energy-saving point of view [18].

CDAE is a non-toxic and eco-friendly extraction method that is being increasingly used for extraction of natural phenolic compounds. However, its use for the extraction of bioactive principles from *H. italicum* has not been described. Having in mind the traditional use of *H. italicum* for dermatological disorders, as well as the variety of phenolic compounds that it contains, the aim of this work was to optimize extraction of phenolics from *H. italicum* aerial parts using CDAE. Anti-elastase and anti-collagenase activity of the prepared extracts were investigated with the aim of obtaining highly active extracts suitable for use in cosmetic products.

## 2. Materials and Methods

### 2.1. Chemicals and Apparatus

For ultrasound-assisted extraction (UAE), a Bandelin SONOREX^®^ Digital 10 P DK 156 BP ultrasonic bath (Berlin, Germany) was used, while a 2mag MIX 15 eco multi-position magnetic stirrer (Munich, Germany) was used for extraction assisted by magnetic stirring. Spectrophotometric determinations were performed using a microplate reader (FLUOstar Omega, BMG Labtech, Ortenberg, Germany). The standards used for the LC-MS analysis, 5-*O*-caffeoylquinic acid and 3,5-*O*-di-caffeoylquinic acid, were purchased from BIOKOM (Janki, Poland). Caffeic acid (CA) was purchased from Carl Roth (Karlsruhe, Germany), while 3-*O*-caffeoylquinic acid was isolated from *Arctium tomentosum* leaves [19], and tiliroside was isolated from aerial parts of *Drymocalis rupestris* (L.) Soják (Rosaceae) [20]. Quantitative Acetonitrile Optima (LC/MS grade) was purchased from Fisher Scientific (Loughborough, UK). Ultrapure water was obtained using the POLWATER DL3-100 system (Kraków, Poland). Formic acid was purchased from Avantor (Gliwice, Poland) to acidify the phases. All standards were of purity ≥ 96%. Other reagents and chemicals were of analytical grade.

### 2.2. Plant Material

*H. italicum* was collected in June 2020 in surroundings of the village Gornja Jagodnja (Zadar county, Croatia; 44°00′28″ N; 15°32′23″ E). The species was authenticated by Suzana Inić. A voucher specimen (HI-2020-6-1) was deposited in the plant collection of the Department of Pharmacognosy, Faculty of Pharmacy and Biochemistry, University of Zagreb (Croatia). Prior to use, fresh aerial parts of *H. italicum*, consisting of leaves, stalks, and flowers, were ground and passed through a sieve of 850 μm mesh size.

### 2.3. Preliminary Solvent Selection

To determine the CD best suited for CDAE, powdered plant material (0.6 g) was placed in a 50 mL Erlenmeyer flask. To the mixture, 10 g of the appropriate CD was added and quickly stirred. Following that, the suspensions were stirred at 25 °C and 300 rpm with a magnetic stirrer. After 24 h, the extracts were filtered using Filtrak Qualitative Folded Filters (Grade 6, 80 g/m^2^, Sartorius, Göttingen, Germany). The detailed extraction conditions are presented in Table 1.

### 2.4. Preliminary Extraction Kinetics

To establish the time needed for ultrasonication procedures, extraction kinetics were determined. Plant material (0.6 g) and HP-β-CD (0.6 mmol) were placed in a 50 mL Erlenmeyer flask. To the mixture, 10 g of water was added, and the contents quickly stirred. The flask was placed in an ultrasonication bath at 30 °C and 360 W ultrasonication power. Samples for determination of total phenols (TP), phenolic acids (TPA), and flavonoids (TF) were taken at 5, 10, 20, 30, 40, 50, and 60 min. The extracts were stored at −20 °C in the dark until use.

### 2.5. Extraction According to the 2-Level Factorial Design

Based on the results of preliminary CDAE, two-level factorial design (2LFD) was performed using five independent variables. Independent variables were as follows (code, minimum value, maximum value): ultrasonication power (X_1_, 144 W to 720 W), temperature (X_2_, 30 °C to 70 °C), lactic acid (LA) content (X_3_, 0% to 2%, *w/w*), herbal material/solvent weight ratio (X_4_, 0.03 to 0.06), and glycerol (GL) content (X_5_, 0% to 5%, *w/w*). Contents of TP, TPA, and TF in the extracts were selected as the dependent variables (responses). For the extraction, fresh powdered plant material in an appropriate amount was placed in a 50 mL Erlenmeyer flask, and 10 g of 0.6 mmol HP-β-CD solution was added and mixed with the appropriate amount of LA and GL. Flask contents were quickly stirred, and the flask placed in an ultrasonication bath at the appropriate ultrasonication power for 30 min. The other extraction variables were in accordance with 2LFD protocol The filtered extracts were stored at −20 °C in the dark until use.

### 2.6. Extraction According to the Box–Behnken Design

After detection of the statistically significant factors, a Box–Behnken design (BBD) was performed using only the dependent variables that statistically significantly affected the dependent variables. The independent variables and their levels were as follows (code, −1, +1): temperature (X_6_, 50 °C, 80 °C), LA content (X_7_, 0% *w:w*, 2% *w/w*) and herbal material weight (X_8_, 0.5 g, 0.9 g). TP, TPA, and TF were selected as the responses. Fresh powdered plant material and 0.6 mmol HP-β-CD were quickly dispersed in 10 g of water, mixed with the appropriate amount of LA and quickly stirred. Following that, the flask was placed in an ultrasonication bath at 144 W ultrasonication power for 30 min. The other extraction variables were in accordance with the BBD protocol. The filtered extracts were stored at −20 °C in the dark until use.

### 2.7. Extraction Optimization

In order to determine the optimal conditions for the preparation of extracts with the desired characteristics, BBD results were analyzed, and CDAE optimization using the response-surface methodology (RSM) and desirability function was performed. The desirability function was used to calculate the optimal extraction conditions for extracting the maximum amounts of TP, TPA, and TF. Two optimized extracts were prepared: OPT 1 (maximum amount of TPA) and OPT 2 (maximum amount of TP and TF). The filtered extracts were stored at −20 °C in the dark until use.

### 2.8. Spectrophotometric Determination of Total Phenolic Content

TP was determined applying the modified Folin–Ciocalteu method, by mixing 80 μL of the extract diluted with water (1:9 *v/v*) with 80 μL of Folin–Ciocalteu reagent and 80 μL of 10% sodium carbonate solution [21]. After 1 h the absorbance at 700 nm was measured and TP concentration was determined from the calibration curve and expressed as μg of gallic acid per mL of extract.

### 2.9. Spectrophotometric Determination of Total Phenolic Acid Content

TPA was determined using a modified method described by Nicolle et al. [21]. To 50 μL of 0.5 M HCl, nitrite-molybdate reagent (prepared using 10 g of sodium nitrite and 10 g of sodium molybdate made up to 100 mL with distilled water), 50 μL of 8.5% NaOH and 100 μL of the extract diluted with water (1:4 *v/v*) were added. TPA was calculated from the calibration curve of caffeic acid and expressed as μg of caffeic acid per mL of extract.

### 2.10. Spectrophotometric Determination of Total Flavonoid Content

TF was determined using a modified colorimetric method [22] by mixing 120 μL extract diluted with methanol (1:3 *v/v*) and 120 μL of 0.2% AlCl_3_ methanolic solution. After 1 h, absorbance at 420 nm was measured. TF was calculated from the calibration curve of quercetin and expressed as μg of quercetin per mL of extract.

### 2.11. LC-MS Analysis

Qualitative analysis was carried out using a 1260 Infinity liquid chromatograph (Agilent, Santa Clara, CA, USA) coupled to a 1290 Infinity photodiode array detector and a 6230 MS-TOF mass spectrometer. Separation of compounds was achieved using mobile phases such as ultrapure water (A) and acetonitrile (B), acidified with 2% (*v*/*v*) formic acid. A 5 μL sample was injected onto a pre-column guarded Zorbax SB-C18 column (250 × 4.6, 5 µm) and eluted using a gradient program (0–23 min, 0–20% B; 23–27 min, 20–22.5% B; 27–28 min, at 24.4% B; 28–30 min, 24.4–26.2% B; 30–50 min, 26.2–30% B; 50–60 min, 30–100% B; 6 min recondition). Solvent flow was 1.2 mL/min. The separation took place at 40 °C. The mass spectrometric conditions were as follows: an electrospray ionization (ESI) source was used in the negative ion mode with drying and sheath gas flow set to 11 L/min and temperature to 350 °C. A nebulizer pressure of 60 psi was used, capillary voltages of 2500 and 1000 V were applied, and the fragmentor was set to 60–320 V. The data were collected in the 120–3000 *m/z* range and analyzed in MassHunter Qualitative 10.0. analysis software. The compounds were characterized based on UV-Vis and MS spectra and the retention time of the standards.

### 2.12. Elastase Inhibitory Activity

For elastase inhibitory activity determination [23], 100 μL of the (1:1 *v/v*) extract solution in Tris-HCl buffer (0.1 M, pH 8.0) was mixed with 25 µL elastase solution (0.05 mg/mL) and left at room temperature for 5 min. Afterwards, a phosphate buffer saline solution of *N*-succinyl-(Ala)_3_-nitroanilide (SANA, 70 µL, 0.41 mg/mL) was added, and the absorbance was measured at 410 nm after an additional 40 min. For the IC_50_ determination, seven different concentrations, prepared using 2-fold dilution, were used. Elastase inhibitory activity (ELAInh) was calculated using Equation (1):(1)$$\mathrm{ELAInh}\;\left(\%\right)=\frac{A_{0}-A_{s}}{A_{0}}\times 100$$
where A_0_ is the absorbance of the negative control, and As is the absorbance of the solution containing the respective extract. Ursolic acid (UA) [24] was applied as the standard elastase inhibitor [24]. Ela IC_50_ values (μL extract/mL) were calculated as the concentration of the extract that inhibited 50% of elastase activity.

### 2.13. Collagenase Inhibitory Activity

To 40 μL of the (1:1 *v/v*) extract solution in Tris-HCl buffer (0.1 M, pH 7.5, 5 mM CaCl_2_, and 1 µM ZnCl_2_), 20 µL of collagenase solution (1 mg/mL dissolved in the same Tris-HCl buffer) was added, and the mixture was incubated at room temperature for 5 min. Following that, gelatin (40 µL, 3.44 mg/mL, dissolved in the same Tris-HCl buffer) was added, and the mixture was incubated at 37 °C. After 40 min, stop reagent, containing 12% (*w/v*) PEG 6000 and 25 mM EDTA, and 90 µL of the ninhydrin reagent were added to the reaction mixture and incubated for 15 min at 80 °C. The ninhydrin reagent for color development was prepared by mixing SnCl_2_ solution (80 mg of SnCl_2_ × 2H_2_O dissolved in 50 mL of 0.2 M, pH 5.0 citrate buffer) with an equal volume of ninhydrin solution (prepared by dissolving 0.5 g of ninhydrin in 10 mL of DMSO). [25]. The inhibition of collagenase (COLInh) was calculated using Equation (2):(2)$$\mathrm{COLInh}\;\left(\%\right)=\frac{A_{0}-A_{s}}{A_{0}}\times 100$$
where A_0_ is the absorbance of the negative control (water), and A_s_ is the absorbance of the respective extract. Gallic acid was used as the positive control [26,27]. COLIC_50_ values (μL extract/mL) were calculated as the concentration of the extract that inhibited 50% of collagenase activity. For the IC_50_ determination, seven different concentrations, prepared using 2-fold dilution, were used.

### 2.14. Statistical Analysis

The measurements were performed in triplicate. The results are presented as mean ± standard deviation. Analyses were performed using one-way ANOVA followed by Tukey post hoc test for comparisons between the extracts. Unless otherwise noted, *p*-values < 0.05 were considered statistically significant. Statistical analyses were performed using GraphPad Prism 8.0.

## 3. Results and Discussion

### 3.1. Extraction Optimization

Extraction conditions can greatly affect the yield and composition of prepared extracts, especially if the target compounds are chemically sensitive [28]. As *H. italicum* is rich in various phenolic metabolites [6] that are prone to oxidation and other types of chemical degradation, selecting the right extraction conditions is of utmost importance [28]. Classical extraction techniques, such as maceration, have a lower yield and longer extraction time than other methods. However, being a mild method that takes place at room temperature, fewer factors may affect the extraction outcome [29]. In order to ensure the constant stirring of plant material particles and their access to the solvent molecules, a similar technique, magnetic stirrer-assisted extraction, was selected as the preliminary extraction method for selection of the most appropriate solvent for the extraction. In this work, the dissolving efficiency of aqueous solutions of three types of CDs: α-CD, HP-β-CD, and HP-γ-CD, tested in up to 8 different concentrations, were compared to the efficiency of classical extraction solvents such as ethanol, water, and mixtures thereof. The amounts of TP, TPA, and TF obtained using different solvents are presented in Figure 1, while the names of the prepared extracts are provided in Table 1.

The results presented in Figure 1 indicate that the addition of CDs resulted in an increase in TP, TPA, and TF in a large majority of solutions. Several CD solutions showed extraction efficiency that matched or even surpassed that of 50% (*w/w*) ethanol, a common solvent for the extraction of phenolic compounds from *H. italicum* [5,6] and other medicinal plants. Contrary to previous reports [5], however, solutions with up to 50% (*w/w*) ethanol content were not the best suited for obtaining extracts with the highest amount of secondary metabolites such as phenolic compounds. In this work, 75% (*w/w*) ethanol was just as good a solvent for the extraction of TP and TPA as 50% (*w/w*) ethanol, while water and other ethanol solutions were less efficient extraction solvents for those groups of metabolites. In general, several HP-β-CD solutions (HPβ600-HPβ1200) were well suited for the CDAE of TP, better than any tested water/ethanol solution. In addition, TF content in 75% (*w/w*) ethanol, the most successful ethanol solution, was statistically equal to the TF in HPβ600-HPβ1200. The findings presented here are in line with numerous studies that demonstrate that β-CDs, and HP-β-CD in particular, are excellently suited for encapsulation of flavonoids such as kaempferol derivatives [30,31] that *H. italicum* is abundant in [14]. Interestingly, the solution of 0.6 mmol of HP-β-CD in 10 g of water was more effective for the dissolution of TPA than the same CD in higher concentrations. This may be surprising because a higher amount of CD molecules in the extracts leads to larger numbers of cavities that may accommodate the active molecules. However, similarly to the swelling of the herbal material with water as described in previous publications [32], solvation of CD molecules may reduce the amount of water molecules that are available for dissolution of target substances. In addition to that, an excessively high CD/solvent ratio leads to unnecessary waste generation. Thus, 0.6 mmol of HP-β-CD dissolved in 10 g of water was chosen for further CDAE optimization.

Further CDAE optimization steps were performed using UAE, a technique with diverse applications in food science. Due to its role in environmental sustainability, ultrasonication is widely used in industries and is referred to as a “green unique technology” [33]. For example, UAE is one of the most practical techniques for large-scale industrial production of algal polysaccharides due to its simplicity, increased yield, high efficiency, and low cost [34]. In addition, high reproducibility, low solvent consumption as well as a fast extraction rate render UAE highly appropriate for the extraction of chemically sensitive bioactive natural compounds, such as polyphenols [35,36]. In order to estimate the optimal duration of UAE with HP-β-CD aqueous solutions, the kinetics of TP, TPA, and TF extraction were studied (Figure 2). As observed, after 30 min the concentration of the target compounds remained constant (TP and TPA) or even slightly decreased. Thus, 30 min was chosen as the extraction time in the following steps.

Plant materials are abundant in various secondary metabolites possessing different pharmacological properties. The task of selectively extracting them with the desired pharmacological properties may be an arduous and lengthy procedure, as their amounts in the extracts depend on numerous extraction parameters such as extraction type, solvent, and temperature, to name a few. To identify the best extraction conditions for the preparation of extracts with desired properties, the work was performed in two steps. The first one was a screening process using 2LFD to select the variables that could significantly affect the extraction. The second step was the BBD, employed to determine the optimum values of the most important independent variables. Such an approach may be used when the factors that could affect a process’s efficiency are so numerous that, without following a statistical approach, it would be difficult to single out the independent variables that, either individually or by cross-effect parameters, affect the process [37], as, for example, in the extraction of glycyrrhizin for determination in herbal drugs [38].

In order to determine the extraction variables that most strongly influence extraction efficacy of phenolic compounds from *H. italicum*, 2LFD was used. The influence of extraction parameters on UAE for green extraction of *H. italicum* phenolics (TP, TPA, and TF) was investigated using six independent variables: HP-β-CD content, the concentrations of GL and LA used as cosolvents, temperature, ultrasonication power, and the amount of herbal material used for the extraction. The results are presented in Table 2.

The employed extraction factors significantly affected the yield of TP, TPA and TF (Table 2). For example, the content of TP ranged from 0.701 ± 0.021 mg/mL to 3.825 ± 0.081 mg/mL in Run 16 and Run 14, respectively, indicating a more than five-fold increase in TP content. TPA content was similarly affected by conditions of CDAE. While Run 6 contained 0.262 ± 0.005 mg/mL, the concentration of TPA in Run 12 was almost six-fold higher, 1.557 ± 0.020 mg/mL. The influence of the extraction parameters on the TF yield was even more pronounced. Run 16 was the extract with the least amount of TF, 0.042 ± 0.003 mg/mL, while Run 12 contained as much as 13 times that amount, 0.563 ± 0.004 mg/mL TF, further stressing the importance of selection of the most successful extraction conditions for maximization of each response.

As the extraction process may be affected by various factors, the importance of optimizing the solvent systems when utilizing CDs as carriers of natural molecules may not be overestimated [39]. Among numerous factors that can influence the efficiency of UAE and other types of extraction, temperature takes a highly important place. High temperature may improve the extraction process by reducing the viscosity of the solvent and increasing the kinetic energy of the molecules in solution. However, high temperature may also cause degradation of the extracted phenolic and other sensitive compounds [40]. Additionally, interaction of phenols with CDs may be an exothermic process, as exemplified in the case of interaction of (E)-piceatannol with HP-β-CD. Thus, high temperature may negatively influence that process [39].

Another factor that may affect the extraction of natural compounds with CDs is the addition of cosolvents. For example, it was shown that the addition of cosolvents, including GL, may improve the solubility of CD-complexed compounds [41]. However, GL addition may also have an opposite effect [42], further illustrating the importance of optimizing the solvent systems when utilizing CDs as drug carriers. In this work, the effects of two cosolvents were investigated. The first one was GL investigated in concentrations up to 5% (*w/w*). GL is a natural, inexpensive, safe, and biodegradable liquid with the additional benefit that it is manufactured from renewable sources, e.g., as a by-product of biodiesel production [43]. It is one of the most widely used ingredients in cosmetic products, often added in creams and lotions, where it acts as a humectant or viscosity-regulating agent [10]. Furthermore, the effects of LA addition were also investigated. Being a weak acid, LA can also affect the pH of the solvent causing the deprotonation of weak acids, thus decreasing their solubility in water but increasing their solubility in nonpolar environments, such as in CD cavities [14]. LA may have either beneficial or adverse effects in the cosmetic product depending on the concentration. Topical application of low concentrations of LA may have a peeling effect and give skin a more youthful appearance. This is achieved by slight disruption of the cohesion of the corneocytes in the skin barrier [44]. However, applications of high concentrations of LA may result in skin irritation [45]. To avoid possible side-effects of a high LA concentration, its content was set up to 2% (*w/w*).

In order to single out the independent variables that significantly affect the responses, Pareto charts were used. Pareto charts depend on the standard deviation to estimate the sampling errors of variables. Each chart depicts two important limits: the Bonferroni limit and *t*-value limit. Variables with coefficients above the Bonferroni limit are considered significant model factors, while the variables that fall between the Bonferroni and the *t*-value limit are considered likely to be significant. Finally, the coefficients below the *t*-value limit are the variables that do not significantly affect the extraction efficiency [46]. Positive and negative effects of independent variables on the dependent ones are indicated by the orange and the blue color on the charts, respectively.

The Pareto chart of the effects of CDAE conditions on the content of target phenolics (Figure 3) showed that the temperature and amount of herbal material used for the extraction were above the Bonferroni limit in all of the graphs and were thus the main factors that positively influenced TP, TPA, and TF. In addition, their combination positively affected the CDAE of TP and TPA, although the intensity of that effect was more dominant in the case of TP. The presence of LA exerted a likely positive influence on TPA content. It has been shown that the presence of weak acids, such as LA, may prevent the alkaline conditions-induced degradation of some flavonoids during the extraction process [47]. Furthermore, LA enhanced the extraction yield and stability of the extracts with phenolic antioxidants of olive leaf [48] and Mediterranean plants [49] in CD-based deep eutectic solvents. It is important to note the possibility that LA may form a eutectic mixture with amino acids present in the plant material, thus enhancing the extraction yield. However, the LA content used in this study (up to 2%) was rather low in comparison to the LA content in other studies that used up to 65% of LA-based eutectic solvent [50]. Thus, even if formation of such a mixture occurred in the extraction medium, its content would have been too low to significantly affect the outcome of the extraction. Unexpectedly, the presence of LA was the main factor that negatively influenced the CDAE of TF. The two other factors, ultrasonication power and GL content, did not affect the extraction efficiency and were thus excluded from the further optimization steps. To verify the validity of CDAE models, the ANOVA analysis (Table 3) was used. It confirmed that the selected models were highly significant (*p* < 0.0001), with high *R*^2^ values (>0.9157), as well as confirming that only the statistically significant effects and the terms supporting the hierarchy were included in the models. Adjusted *R*^2^ and predicted *R*^2^ were in reasonable agreement, further confirming the validity of the models.

In order to determine the extraction conditions best suited for extraction of phenolic metabolites from *H. italicum*, BBD, followed by response surface methodology (RSM), was used. RSM is a collection of mathematical and statistical techniques aimed at creating a functional relationship and empirical model between the response(s) of interest and several associated input variables. This allows for optimization of the processes, including extraction, and selection of the conditions most appropriate for achievement of process goals, e.g., maximization of the extraction yield of target compounds.

BBD followed by RSM is frequently applied for optimization of UAE of phenolic compounds from various natural sources such as apple pomace [51], olive leaf [52], and yellow bell pepper waste [53]. However, its use for optimization of UAE of natural phenols by CDs is not so widespread. The few examples include the extraction of phenolic compounds from alfalfa [54] and common bird’s-foot trefoil [55]. In order to fine-tune CDAE and maximize the yield of TP, TPA and TF in this work, only the variables that significantly affected the target phenolics in the 2LFD were selected as the independent variables in the BBD: temperature, LA content, and the amount of herbal material taken into CDAE. The contents of TP, TPA and TF in the *H. italicum* extracts prepared according to the BBD are presented in Table 4. The amount of extracted TP differed among extracts, from 2.248 ± 0.013 mg/mL recorded in Run 11 to 4.185 ± 0.049 mg/mL in Run 7. The lowest TPA content (1.170 ± 0.041 mg/mL) was obtained in Run 12, while the lowest TPA was displayed by Run 8 (2.171 ± 0.117 mg/mL). The amount of TF ranged from 0.171 ± 0.012 mg/mL (Run 5) to 0.556 ± 0.006 mg/mL (Run 7). It is important to note that the highest TP and TPA contents recorded in BBD were significantly higher than those recorded in 2LFD (3.825 ± 0.081 mg/mL and 1.557 ± 0.020 mg/mL, respectively). On the other hand, the highest TF recorded in BBD was slightly lower than that of the 2LFD (0.563 ± 0.004 mg/mL). However, that difference did not reach statistical significance (*t*-test, *p* < 0.05).

In order to explore the relationships between the dependent and independent variables, regression analysis was performed, and the relationship between variables was represented by the corresponding models. Equations (3)–(5) show the fitted model equations with statistically significant factors (*p* < 0.05) marked with bold font and an asterisk (*).
TP (mg/mL) = 3.03 + 0.27 × ***X_6_** − 0.24 × ***X_7_** + 0.63 × ***X_8_** + 0.12 × X_6_ × X_7_ + 0.21 × ***X_6_** × **X_8_** − 0.15 × X_7_ × X_8_(3)
TPA (mg/mL) = 1.66 + 0.15 × ***X_6_** + 0.09 × X_7_ + 0.35 × ***X_8_**(4)
TF (mg/mL) = 0.31 + 0.001 × X_6_^2^ + 0.048 × ***X_7_^2^** + 0.004 × X_8_^2^ + 0.021 × X_6_ × X_7_ + 0.002 × X_6_ × X_8_ − 0.032 × ***X_7_** × **X_8_** + 0.030 × ***X_6_** − 0.120 × ***X_7_** + 0.070 × ***X_8_**(5)

As may be observed from Equation (3), TP was best described with a two-factor-interaction model. All of the selected variables in the applied range significantly influenced the TP yield. Temperature and herbal material content positively influenced TP content both as linear terms and in the form of their interaction. The effect of LA content was also linear but negative. On the other hand, as may be seen in Equation (4), the independent variables linearly and positively affected TPA content, while the relationship between TF and the dependent variables was best described using a polynomial quadratic equation (Equation (5)). Similarly to TP and TPA extraction, temperature and weight of herbal material positively affected the CDAE outcome. Here, the mixed role of LA in the CDAE of TF could be observed, because it positively affected the TF content as a quadratic term, and negatively as a linear term. In addition, the interaction of LA and temperature also adversely affected the extraction efficiency.

To determine the statistical significance of the obtained models, ANOVA was performed, and *p*-values and *F*-test were calculated. As may be observed in Table 5, the calculated *F*-values of both models were higher than 10, while the *p*-values for the models were lower than 0.001. This confirmed the significance of the selected models, as well as their suitability for the interpretation of the experimental data. The determination coefficients (*R*^2^) for CDAE of TP and TF were rather high (0.9519 and 0.9774, respectively), while *R*^2^ for TPA was somewhat lower but still satisfactory (0.7168), confirming that the observed values were well described by this model. The predicted *R*^2^ values for all of the models were in good agreement with the adjusted ones, additionally proving the value of the models.

The values of independent variables most appropriate for the preparation of extracts with the highest TP, TPA, and TF content were calculated. The CDAE conditions applied for preparation of two optimized extracts, OPT 1 (having the highest TPA content) and OPT 2 (highest TP and TF content), were calculated and presented in Table 6. It may be observed that the extraction conditions were fairly similar, the only difference being the LA content, which was nearing a maximum in the TPA extraction. As LA was not a significant factor for TPA extraction in BBD, its addition could have been omitted. However, the 2LFD identified LA content as a possibly significant factor. Therefore, to explore the difference in the composition of the extracts prepared using different LA concentrations, the extract was prepared according to the calculated conditions and the quantity of TP, TPA and TF determined. The amounts of the target phenolics in the prepared extracts were significantly higher (*t*-test, *p* < 0.05) than their respective highest values recorded in BBD, confirming the importance of careful selection of CDAE conditions for the best extraction outcomes. The predicted values of the responses in the prepared extracts were in good accordance with the predicted ones with response deviation being in range of ±6%. The obtained results further confirmed the suitability of the calculated models to adequately describe and predict extraction outcomes.

### 3.2. LC-MS Analysis of the Extracts

Previous studies on various extracts and preparations have shown that *H. italicum* is a rich source of diverse phenolic compounds such as phenolic acids (hydroxybenzoic and hydroxycinnamic acids), hydroxycinnamic esters, coumarins, flavonoids (flavones, flavonols, flavanones, flavanols), flavonoid ethers, esters and glycosides, and acetophenones as well as associates of those classes. Published data on the identification of these compounds by spectroscopic methods largely facilitate explanation of the chemical composition of additional extracts and preparations using chromatographic methods [6,56,57].

UV chromatograms of the prepared optimized extracts (OPT 1 and OPT 2) recorded at 270 nm and 365 nm revealed the presence of 25 compounds (Figure 4). After interpretation of the fragmentation pattern from the collected mass spectra of two optimized *H. italicum* extracts, 22 phenolic compounds were tentatively identified, while the identity of the three remaining peaks remained unknown. Of these, five compounds could be unambiguously identified with a reference standard. The other 20 structures were tentatively identified by comparison with the literature and/or confirmed by exact mass match and isotopic fragment pattern comparison. Detailed information on the identified compounds is available in Table 7 and Figure 4, where the compounds are listed in order of elution from the column and numbered accordingly. The same numbering system is maintained throughout the text.

Among the (tentatively) identified structures (Table 7), the most prevalent were the derivatives of hydroxycinnamic acid (**3**–**6**, **11**–**15**, **19**), mostly derivatives of caffeic acid, especially caffeoylquinic acid derivatives. Flavonols were another highly represented group, with the derivatives of aglycone quercetin (**8**,**10**,**16**,**17**) being the most represented. The other aglycones included myricetin (**7**), isorhamnetin (**9**), and kaempferol (**18**). In addition, two phloroglucinol α-pyrones were also detected—arzanol (**23**) and 3-methylarzanol (**24**). As depicted in Figure 4, the most prominent peak in both chromatograms was that of 3,5-*O*-dicaffeoylquinic acid (**12**). Additionally, chromatograms of both extracts recorded at 270 nm revealed the significant presence of an unidentified quercetin coumaroylglucoside isomer (**17**). It is possible that some of the molecules identified in the extracts form inclusion complexes with CD molecules. For example, it has been demonstrated that caffeic acid forms an inclusion complex with HPβCD with a 1:1 stoichiometry [58]. In addition, HPβCD may form complexes with some of the flavonoids and their derivatives present in the extracts, such as myricetin [59], quercetin, and quercetin glycosides such as rutin [14,60], thus enhancing their solubility and stability.

The addition of a small amount of LA did not significantly influence the composition of the extracts. One of the differences was that the peak belonging to myricetin 3-*O*-glucoside (**7**) was higher in OPT 2, the extract prepared without LA. On the other hand, the arzanol (**23**) and 3-methylarzanol (**24**) peaks, while present in both extracts, were more pronounced in OPT 1, prepared using 1.95% (*w*/*w*) LA. Arzanol displays antioxidant, anti-HIV, as well as a potent anti-inflammatory activity that makes it a candidate for development of treatments for autoimmune diseases and cancer [7]. Thus, the observed increase in the extraction efficiency caused by addition of small amounts of LA may have important implications for the pharmaceutical and nutraceutical industries.

Two *H. italicum* extracts contained numerous phenolic antioxidants. Flavonoids in particular display diverse biological activity that makes them valuable ingredients in cosmetic and dermatologic products. In addition to their antioxidant activity, they act as photoprotectants and anti-inflammatory, depigmentation, and anti-aging agents [61]. Several studies have shown the anti-aging and skin-protecting potential of quercetin, the flavonoid whose derivatives were the most represented in the prepared extracts. Quercetin promotes cutaneous wound healing [62], suppresses ultraviolet (UV) radiation-induced matrix metalloproteinase-1 (MMP 1) and cyclooxygenase-2 expression, inhibits UV radiation-mediated collagen degradation in human skin tissues, and prevents UV radiation-induced photoaging in human skin by directly targeting Janus kinase (2 JAK_2_) and protein kinase C-δ (PKCδ) [63]. Another group of phenolic compounds highly represented in the OPT 1 and OPT 2 were hydroxycinnamic acids. Similarly to flavonoids, they act as photoprotective, anti-inflammatory, and depigmenting agents. Their anti-aging activity is mostly attributed to their ability to inhibit the activity of matrix-degrading enzymes such as collagenase [64]. For example, one in vitro study on 3,5-*O*-dicaffeoylquinic acid isolated from *Artemisia lavandulaefolia*, found that it exhibited anti-rosacea effects [65]. Additionally, 3,5-*O*-dicaffeoylquinic acid demonstrated an anti-photoaging effect against skin damage induced by radiation in the UVA region (400–315 nm) and protected human dermal fibroblasts in vitro [66]. As 3,5-*O*-dicaffeoylquinic acid was the most prevalent compound in both extracts, this confirmed the potential of the prepared extracts for use in cosmetics. Finally, the phloroglucinol α-pyrone arzanol inhibits eicosanoid biosynthesis and exhibits strong anti-inflammatory efficacy in vivo [67].

**Table 7 metabolites-13-00257-t007:** The spectroscopic data for the compounds observed in *H. italicum* extracts obtained with LC-PDA-ESI-MS.

No	Rt [min]	λ Max [nm]	Obs ^A^	Diff [ppm]	Formula	*m/z* ESI-	Compound Name	Identification Type ^B^
1	9.40	260, 292	153.02	−2.14	C_7_H_6_O_4_	109, 153	Dihydroxybenzoic acid derivatives	[6], MS
2	10.14	258, 290, 328	327.07	−1.43	C_14_H_16_O_9_	165, 327	Hydroxyphtalide glucoside isomer	[6], MS
3	11.24	244, 294 324	353.09	−0.95	C_16_H_18_O_9_	179, 191, 353	5-*O*-caffeoylquinic acid	ST, [6,68], MS
4	14.83	244, 294, 324	353.09	−2.03	C_16_H_18_O_9_	191, 353	3-*O*-caffeoylquinic acid	ST, [6,56,68], MS
5	16.06	246, 290, 324	179.03	−4.59	C_9_H_8_O_4_	179	Caffeic acid	ST, [6,56], MS
6	20.29	248, 282, 324	367.10	−0.67	C_17_H_20_O_9_	191, 367	5-*O*-feruloyloquinic acid	[6,68], MS
7	20.53	278, 340, 385	479.08	−0.89	C_21_H_20_O_13_	477	Myricetin 3-*O*-glucoside	[6], MS
8	21.37	256, 282, 338	463.03	−5.11	C_21_H_20_O_12_	300, 463	Quercetin *O*-hexoside isomer	[6], MS
9	23.82	254, 330	521.09	−7.41	C_23_H_22_O_14_	521	Isorhamnetin *O*-hexoside	[57], MS
10	24.22	255, 287, 345	463.08	−9.49	C_21_H_20_O_12_	300, 463	Quercetin O-hexoside isomer	[6], MS
11	25.70	250, 295, 325	515.12	−8.19	C_25_H_24_O_12_	515, 353	3,4-*O*-Dicaffeoylquinic acid isomer	[57,68], MS
12	26.49	250, 295, 325	515.11	−9	C_25_H_24_O_12_	191, 353, 515	3,5-*O*-Dicaffeoylquinic acid	ST, [6,57,68], MS
13	26.96	250, 295, 328	695.12	−8	C_33_H_28_O_17_	209, 371, 515, 695	Tricaffeoyl hexaric acid	[6], MS
14	28,44	250, 295, 328	515.12	−7.12	C_25_H_24_O_12_	191, 353, 515	4,5-*O*-Dicaffeoylquinic acid isomer	[57,68], MS
15	32.06	250, 295, 328	515.12	−6.61	C_25_H_24_O_12_	353, 515	Dicaffeoylquinic acid isomer	MS
16	32.48	255, 270, 315, 360	609.12	−8.01	C_30_H_26_O_14_	300, 463, 609	Quercetin coumaroylglucoside isomer	[6], MS
17	33.12	255, 290, 355	609.12	−6.44	C_30_H_26_O_14_	300, 463, 609	Quercetin coumaroylglucoside isomer	[6], MS
18	35.86	252, 296, 355	593.13	−6.81	C_30_H_26_O_13_	284, 593	Tiliroside	ST, [6,56,57], MS
19	36.04	252, 296, 325	677.15	−5.41	C_34_H_30_O_15_	515, 677	Tricaffeoylquinic acid	[6], MS
20	37.26	252, 276	629.15	2.47	C_37_H_26_O_10_	629	Quinic acid derivatives	MS
21	55.22	250, 280	435.13	5.74	C_28_H_20_O_5_	435	Unknown	MS
22	57.29	280	417.15	5.11	C_29_H_22_O_3_	417	Unknown	MS
23	60.09	252, 292, 360	401,17	−2	C_22_H_26_O_7_	401	Arzanol	[6], MS
24	60.62	294, 360	415,18	−0.52	C_23_H_28_O_7_	415	3-Methylarzanol	[6], MS
25	61.40	294, 350	429.20	−2.64	C_17_H_34_O_12_	429	unknown	MS

Obs = observed; ^A^ = exact mass registered in [M-H]^−^; ^B^ = confirmed by comparison with the literature; bold = most abundant; ST = confirmed with reference substance; MS = confirmed by exact mass match, isotopic fragment pattern comparison.

### 3.3. Anti-Elastase and Anti-Collagenase Activity of the Optimized Extracts

In order to test the anti-aging potential of the prepared extracts, their activity against collagenase and elastase was investigated. Enzyme-inhibiting activity of the extracts is displayed in Figure 5. Positive controls (ursolic and gallic acid) were also evaluated and their activity displayed. However, it is important to have in mind that the activity of the extracts and the activity of the standards is expressed in different measurement units, μL extract/mL and μg/mL, respectively. Therefore, their activities may not be directly compared. However, for comparison purposes, it is possible to regard the activity of the standards as volume equivalents of 1 mg/mL solutions.

Elastin is an important matrix protein responsible for maintaining the mechanical properties of the skin [69]. The degradation of elastin is produced by the activity of the enzyme elastase, a process that is directly related to skin aging and oxidative stress [70]. Elastase is a protease whose increased activity is related to various ailments and skin problems, such as psoriasis and prolonged wound healing. In addition, increased elastase activity and consequential elastin degradation are among the most important causes of skin elasticity loss, premature skin aging, and the formation of wrinkles [71]. Therefore, it is not surprising that numerous clinical trials have confirmed that the inhibition of elastase is related to the prevention of structural damage to the extracellular matrix and, accordingly, to photoaging protection [71,72]. The elastase inhibitory activity of OPT 1, OPT 2, and positive control is presented in Figure 5. While OPT 2 was a stronger elastase inhibitor than OPT 1, both extracts demonstrated excellent anti-elastase potential, statistically stronger than the activity of the tested ursolic acid solution. This activity may be connected to the presence of 3,5-dicaffeoylquinic acid in the extracts. In a study that used 3,5-dicaffeoylquinic acid isolated from *Ilex kaushue*, it was found that it exerted a potent and selective inhibitory effect on human neutrophil elastase [73]. In addition, quercetin also produces significant anti-elastase activity [74], while *H. italicum* subsp. *italicum* essential oil as well as its constituents *α*-pinene and limonene display in vitro anti-elastase activity [8]. However, as OPT 1 and OPT 2 were water-based extracts, it is not clear how much the essential oil components can contribute to the effect observed in this study.

Collagen is the dominant protein in matrix and constitutes about 80–90% of the dermis [75]. While elastin fibers contribute to the extensibility and reversible recoil of the skin, collagen contributes to the tensile strength and stability of skin tissue [76]. Collagenases are the enzymes active in the extracellular matrix that contribute to the degradation of collagen [77]. For example, interstitial collagenase (MMP 1) is able to hydrolyze type I collagen, the major component of the dermis, and thus plays a crucial role in the disorganization and progressive degeneration of dermal extracellular matrix [78]. With aging and various external influences (UV radiation), its activity increases and leads to the formation of wrinkles and loss of skin tone [77]. As shown in Figure 5, OPT 1 was the stronger collagenase inhibitor, yet both extracts demonstrated a statistically significant superior inhibitory effect compared to the positive control, gallic acid. Similarly to the anti-elastase activity, *H. italicum* subsp. *italicum* essential oil and the *α*-pinene and limonene that it contains displayed in vitro anti-collagenase activity [8]. However, as the essential oil components are highly lipophilic, it is not likely that their concentration in the CD-aqueous extracts would be sufficient to be solely responsible for the observed effects. It is interesting to note that 3,5-*O*-dicaffeoylquinic acid decreases interstitial collagenase production and increases type I collagen production in UVA-damaged human dermal fibroblasts [66].

## 4. Conclusions

*H. italicum* is a plant traditionally used for skin-related disorders that is becoming an increasingly popular ingredient in cosmetic products. In this work, the influence of CDAE conditions on the content of phenolic compounds was investigated. Among the tested CDs, HP-β-CD was the most efficacious for extraction of target phenolics and used to prepare two extracts with the highest TPA, TP, and TF. The main metabolite in both extracts was 3,5-*O*-dicaffeoylquinic acid. It was found that the addition of LA greatly contributed to the extraction of arzanol, a metabolite with pronounced anti-inflammatory properties. Anti-elastase and anti-collagenase activities of the prepared extracts surpassed those of positive controls, ursolic and gallic acid. The extracts were prepared using UAE, a green extraction technique used for industrial applications in food and medicinal plant extractions. The prepared extracts are appropriate for use in cosmetic products without solvent evaporation, saving thus the time and energy necessary for product manufacture. However, prior to their inclusion in products for human use, further research on the biological properties and stability of the prepared extracts should be performed.

## Figures and Tables

**Figure 1 metabolites-13-00257-f001:**
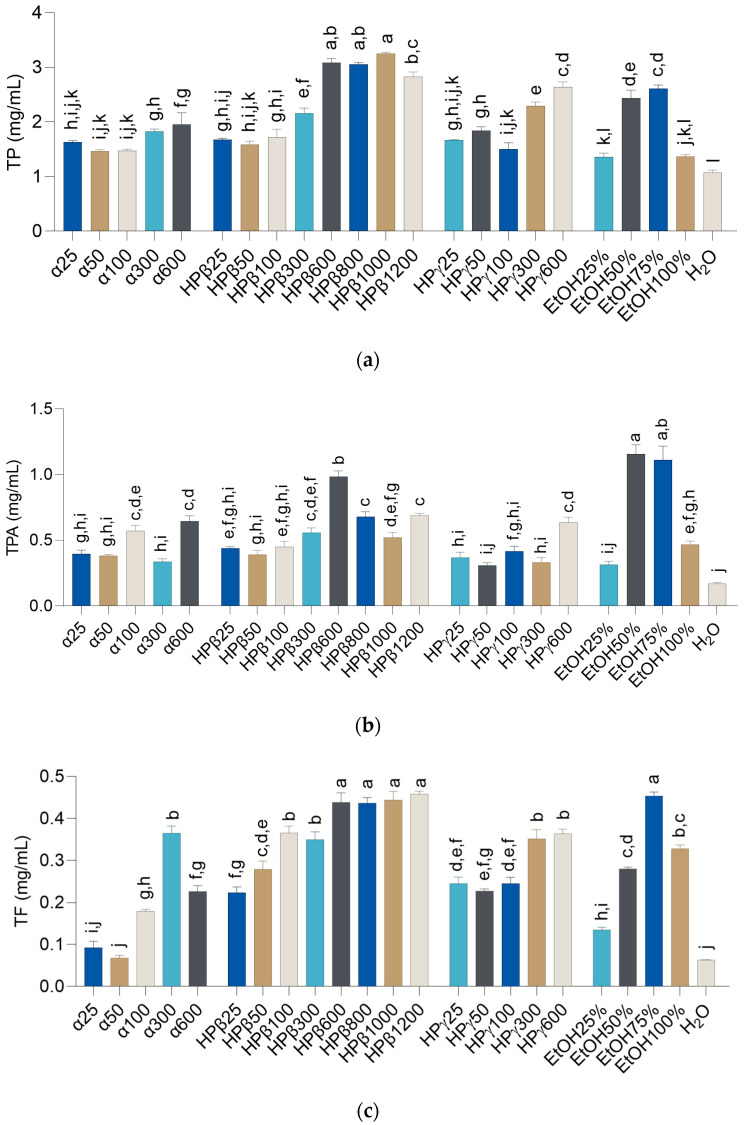
Content of total (**a**) phenolic compounds (TP), (**b**) phenolic acids (TPA), and (**c**) flavonoids (TF) in preliminary *H. italicum* extracts. The abbreviated extract names are presented in Table 1. The experiments were performed in triplicate. The amounts are shown as mean ± SD. ^a–l^ = differences between the extracts within a column (Tukey post-test, *p* < 0.05). Values not connected with the same letter are statistically different.

**Figure 2 metabolites-13-00257-f002:**
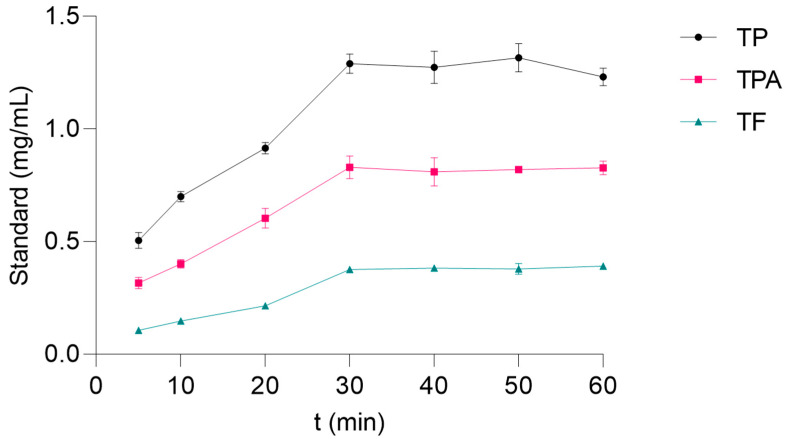
Kinetic plots of content of total phenolics (TP), phenolic acids (TPA), and flavonoids (TF) extracted from *H. italicum* using a solution of 0.6 mmol of HP-β-CD in 10 g of water. The experiments were performed in triplicate. The amounts are shown as mean ± SD. Standards: gallic acid (TP), caffeic acid (TPA), and quercetin (TF).

**Figure 3 metabolites-13-00257-f003:**
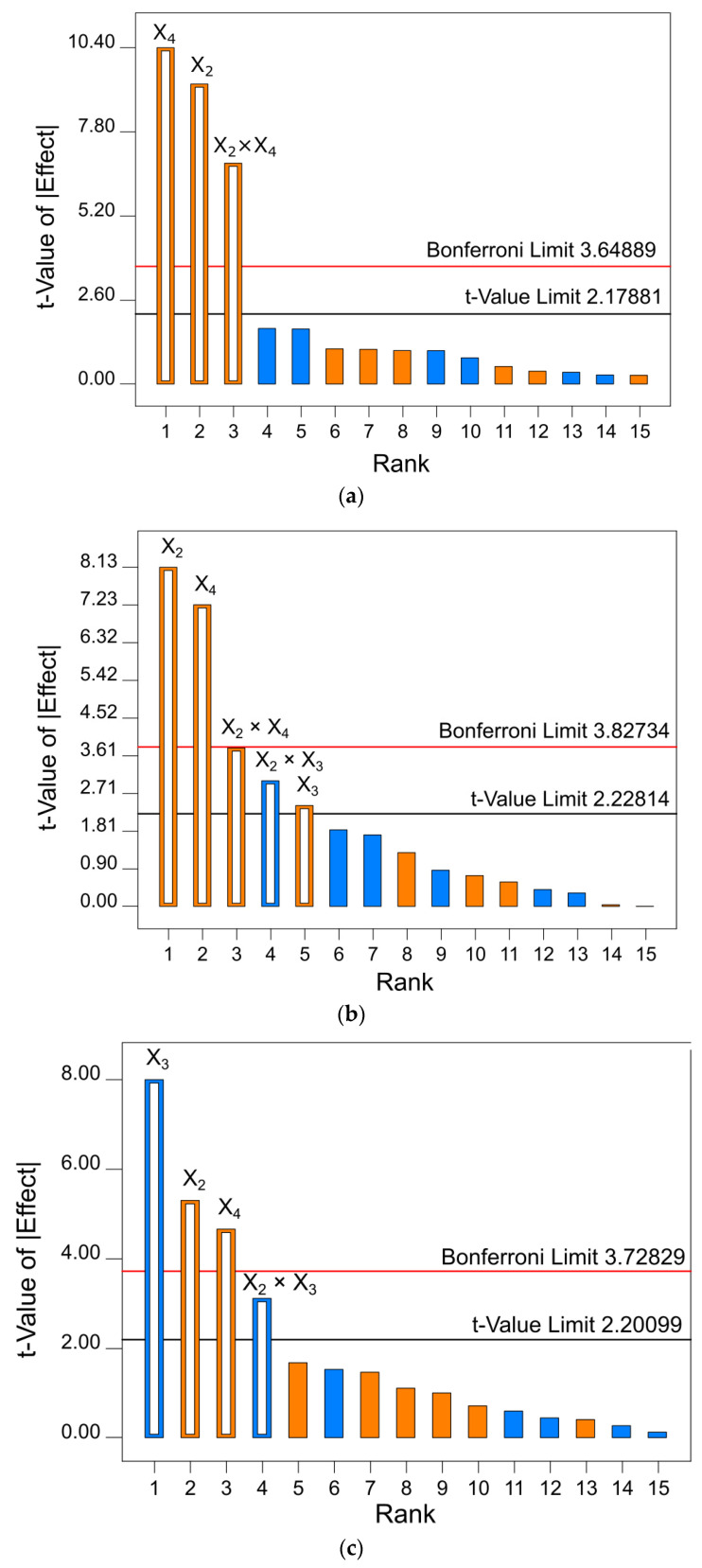
Pareto charts for the selected models of total (**a**) phenolic compounds, (**b**) phenolic acids, and (**c**) flavonoids extraction. Independent variables: X_1_ = ultrasonication power, X_2_ = temperature, X_3_ = lactic acid content, X_4_ = herbal material/solvent weight ratio, X_5_ = glycerol content. Blue color on the chart (**a**) indicates a negative, and orange color a positive, effect of independent variables.

**Figure 4 metabolites-13-00257-f004:**
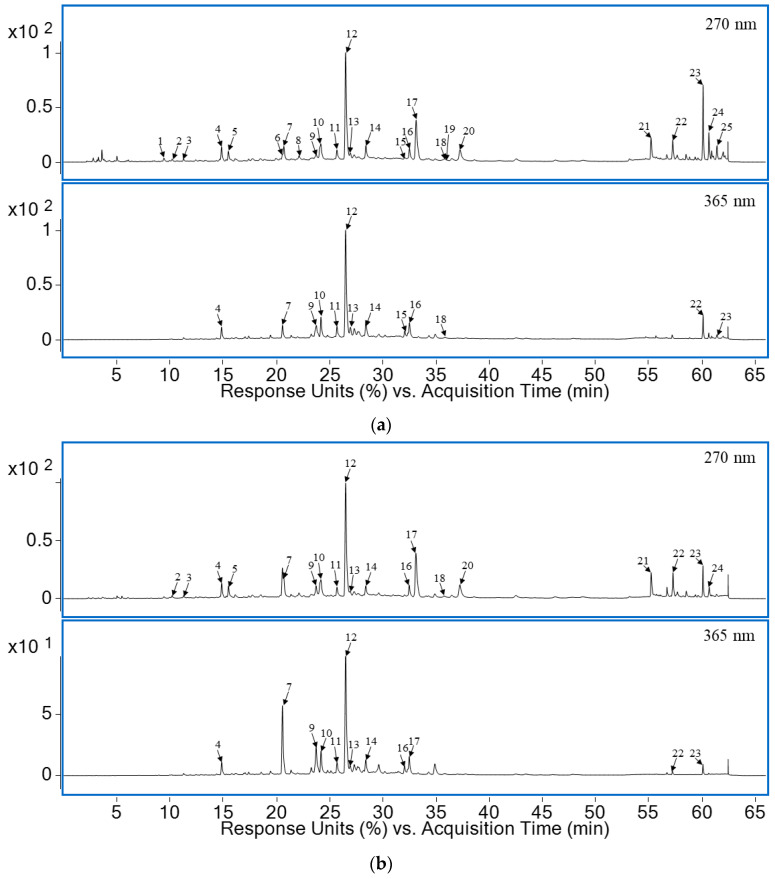
UV chromatograms of (**a**) OPT 1 and (**b**) OPT 2 extracts obtained by LC-PDA-MS. The compound labels are explained in Table 7.

**Figure 5 metabolites-13-00257-f005:**
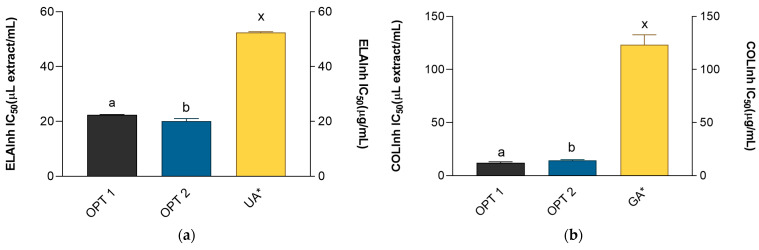
Inhibitory activity of the extracts and corresponding positive controls: ursolic acid (UA) and gallic acid (GA) on (**a**) elastase and (**b**) collagenase. The activities are shown as IC_50_ values ± SD. ^a, b^ = differences between the extracts within a column (*t*-test, *p* < 0.05). ^x^ = differences with the positive control (Dunnett’s post-test, *p* < 0.05). Columns not connected with the same letter are statistically different. Asterisk indicates that the unit is placed at the right ordinate.

**Table 1 metabolites-13-00257-t001:** Conditions for preliminary extraction procedure using either cyclodextrins (CDs) dissolved in water (10 g) or mixtures of ethanol and/or water (10 g) stirred at 25 °C and 300 rpm for 24 h.

Extract	CD Type *	n (CD) (mmol)
α25	α-CD	0.025
α50	α-CD	0.050
α100	α-CD	0.100
α300	α-CD	0.300
α600	α-CD	0.600
HPβ25	HP-β-CD	0.025
HPβ50	HP-β-CD	0.050
HPβ100	HP-β-CD	0.100
HPβ300	HP-β-CD	0.300
HPβ600	HP-β-CD	0.600
HPβ800	HP-β-CD	0.800
HPβ1000	HP-β-CD	1.000
HPβ1200	HP-β-CD	1.200
HPγ25	HP-γ-CD	0.025
HPγ50	HP-γ-CD	0.050
HPγ100	HP-γ-CD	0.100
HPγ300	HP-γ-CD	0.300
HPγ600	HP-γ-CD	0.600
**Extract**	**Co-solvent ****	**Co-solvent concentration (%, *w*/*w)***
EtOH25%	Ethanol	25
EtOH50%	Ethanol	50
EtOH75%	Ethanol	75
EtOH100%	Ethanol	100
H_2_O	-	0

n (CD) = amount of the cyclodextrin; α-CD = α-cyclodextrin; HP-β-CD = (2-hydroxypropyl)-β-cyclodextrin; HP-γ-CD = (2-hydroxypropyl)-γ-cyclodextrin. * = dissolved in water; ** mixed with water.

**Table 2 metabolites-13-00257-t002:** Independent variables in the two-level factorial design and content of total phenols (TP), phenolic acids (TPA), and flavonoids (TF) in the extracts.

Standard	Run	X_1_ (W)	X_2_ (°C)	X_3_ (%, *w/w*)	X_4_	X_5_ (%, *w/w*)	TP (mg/mL)	TPA (mg/mL)	TF (mg/mL)
1	1	144.00	30.0	0.00	0.03	5.00	0.883 ± 0.028	0.276 ± 0.010	0.246 ± 0.016
4	2	720.00	70.0	0.00	0.03	5.00	1.102 ± 0.035	0.532 ± 0.034	0.420 ± 0.014
8	3	720.00	70.0	2.00	0.03	0.00	1.237 ± 0.017	0.753 ± 0.035	0.134 ± 0.006
2	4	720.00	30.0	0.00	0.03	0.00	1.271 ± 0.017	0.683 ± 0.011	0.191 ± 0.008
14	5	720.00	30.0	2.00	0.06	0.00	2.864 ± 0.009	1.364 ± 0.039	0.282 ± 0.018
7	6	144.00	70.0	2.00	0.03	5.00	0.942 ± 0.016	0.262 ± 0.005	0.286 ± 0.014
13	7	144.00	30.0	2.00	0.06	5.00	1.441 ± 0.028	0.825 ± 0.016	0.245 ± 0.008
16	8	720.00	70.0	2.00	0.06	5.00	3.353 ± 0.134	1.313 ± 0.056	0.328 ± 0.009
9	9	144.00	30.0	0.00	0.06	0.00	1.334 ± 0.046	0.401 ± 0.004	0.301 ± 0.026
10	10	720.00	30.0	0.00	0.06	5.00	1.353 ± 0.017	0.756 ± 0.034	0.444 ± 0.018
15	11	144.00	70.0	2.00	0.06	0.00	1.530 ± 0.042	0.894 ± 0.007	0.230 ± 0.007
12	12	720.00	70.0	0.00	0.06	0.00	3.754 ± 0.108	1.557 ± 0.020	0.563 ± 0.004
5	13	144.00	30.0	2.00	0.03	0.00	1.184 ± 0.008	0.721 ± 0.002	0.215 ± 0.007
6	14	720.00	30.0	2.00	0.03	5.00	3.825 ± 0.081	1.426 ± 0.066	0.545 ± 0.021
3	15	144.00	70.0	0.00	0.03	0.00	1.233 ± 0.046	0.363 ± 0.004	0.354 ± 0.020
11	16	144.00	70.0	0.00	0.06	5.00	0.701 ± 0.021	0.310 ± 0.006	0.042 ± 0.003

X_1_ = ultrasonication power, X_2_ = temperature, X_3_ = lactic acid content, X_4_ = herbal material/solvent weight ratio, X_5_ = glycerol content. The experiments were performed in triplicate. The amounts are shown as mean ± SD.

**Table 3 metabolites-13-00257-t003:** Coefficients of determination (*R*^2^) and *p*-value of ANOVA analysis for the models of extraction of total phenols (TP), total phenolic acids (TPA), and total flavonoids (TF).

Dependent Variable	*R* ^2^	*R* ^2^ _P_	*R* ^2^ _A_	Model *p*-Value
TP	0.9526	0.9407	0.9157	<0.0001
TPA	0.9366	0.9050	0.8378	<0.0001
TF	0.9183	0.8886	0.8272	<0.0001

*R*^2^_A_ = adjusted *R*^2^; *R*^2^_P_ = predicted *R*^2^.

**Table 4 metabolites-13-00257-t004:** Extraction conditions and content of total phenols (TP), phenolic acids (TPA), and flavonoids (TF) in the extracts prepared according to the Box–Behnken design.

Standard	Run	X_6_	X_7_	X_8_	TP	TPA	TF
		(°C)	(%, *w/w*)	(g)	(mg/mL)	(mg/mL)	(mg/mL)
10	1	65.00	2.00	0.50	2.354 ± 0.013	1.559 ± 0.025	0.217 ± 0.003
1	2	50.00	0.00	0.70	3.083 ± 0.092	1.174 ± 0.016	0.478 ± 0.017
4	3	80.00	2.00	0.70	3.117 ± 0.064	1.758 ± 0.143	0.277 ± 0.002
15	4	65.00	1.00	0.70	3.277 ± 0.104	1.872 ± 0.062	0.347 ± 0.014
3	5	50.00	2.00	0.70	2.408 ± 0.110	1.555 ± 0.039	0.171 ± 0.012
6	6	80.00	1.00	0.50	2.435 ± 0.036	1.274 ± 0.037	0.260 ± 0.014
11	7	65.00	0.00	0.90	4.185 ± 0.049	1.892 ± 0.082	0.556 ± 0.006
8	8	80.00	1.00	0.90	4.052 ± 0.005	2.171 ± 0.117	0.408 ± 0.008
7	9	50.00	1.00	0.90	3.009 ± 0.161	1.962 ± 0.046	0.348 ± 0.013
14	10	65.00	1.00	0.70	2.960 ± 0.096	1.825 ± 0.042	0.310 ± 0.019
5	11	50.00	1.00	0.50	2.248 ± 0.013	1.287 ± 0.026	0.206 ± 0.002
9	12	65.00	0.00	0.50	2.563 ± 0.053	1.170 ± 0.041	0.357 ± 0.004
13	13	65.00	1.00	0.70	3.117 ± 0.111	1.655 ± 0.074	0.304 ± 0.001
16	14	65.00	1.00	0.70	3.121 ± 0.047	1.709 ± 0.005	0.299 ± 0.005
2	15	80.00	0.00	0.70	3.330 ± 0.017	1.975 ± 0.027	0.502 ± 0.013
12	16	65.00	2.00	0.90	3.380 ± 0.071	2.054 ± 0.202	0.288 ± 0.004
17	17	65.00	1.00	0.70	2.810 ± 0.053	1.252 ± 0.014	0.289 ± 0.013

X_6_ = temperature, X_7_ = lactic acid content, X_8_ = weight of herbal material; TP = total phenolics content, TPA = total phenolic acids content, TF = total flavonoids content. For extraction, herbal material was dispersed in 10 g solvent containing 0.6 mmol HP-β-CD and extracted for 30 min.

**Table 5 metabolites-13-00257-t005:** Analysis of variance (ANOVA) and coefficients of determination (*R*^2^) for the fitted model equations of contents of total phenols (TP), phenolic acids (TPA), and flavonoids (TF).

	TP				
*R* ^2^	*R*^2^ = 0.9519, *R*^2^_P_ = 0.9230, *R*^2^_A_ = 0.8632				
Source	SS	DF	MS	*F* Value	*p*-value
Model	4.53	6	0.76	32.97	<0.0001
Lack of Fit	0.10	6	0.017	0.54	0.7621
Pure Error	0.13	4	0.032		
	**TPA**				
*R* ^2^	*R*^2^ = 0.7168, *R*^2^_P_ = 0.6514, *R*^2^_A_ = 0.5444				
Source	SS	DF	MS	*F* Value	*p*-value
Model	1.22	3	0.41	10.97	0.0007
Lack of Fit	0.24	9	0.027	0.44	0.8586
Pure Error	0.24	4	0.060		
	**TF**				
*R* ^2^	*R*^2^ = 0.9774, *R*^2^_P_ = 0.9483, *R*^2^_A_ = 0.8015				
Source	SS	DF	MS	*F* Value	*p*-value
Model	0.17	9	0.019	33.63	<0.0001
Lack of Fit	0.002	3	0.001	1.33	0.3812
Pure Error	0.002	4	0.001		

SS = sum of squares, DF = degrees of freedom, MS = mean square, *R*^2^_A_ = adjusted *R*^2^; *R*^2^_P_ = predicted *R*^2^.

**Table 6 metabolites-13-00257-t006:** Predicted and observed response values for the extracts prepared in optimal conditions.

Extract Name	Optimized Response (mg/mL)	Response Aim	X_6_ (°C)	X_7_ (%, *w/w*)	X_8_ (g)	Predicted Response Value (mg/mL)	Observed Response Value (mg/mL)	RD (%)
OPT 1	TPA	maximized	80	1.95	0.89	2.23	2.11 ± 0.07	−5.67
OPT 2	TP and TF	minimized	80	0	0.89	4.28 and 0.63	4.38 ± 0.04 and 0.60 ± 0.02	2.28 and −5.00

X_6_ = temperature, X_7_ = lactic acid content, X_8_ = drug weight, RD = response deviation calculated as (Observed–Predicted)/Predicted × 100, TP = total phenolic content, TPA = total phenolic acid content, TF = total flavonoid content.

## Data Availability

Data is not publicly available due to privacy or ethical restrictions.

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
