# Peer review of "Optimization of Cyclodextrin-Assisted Extraction of Phenolics from Helichrysum italicum for Preparation of Extracts with Anti-Elastase and Anti-Collagenase Properties"

_metabolites, 2023, doi:10.3390/metabo13020257_

Round 1

Reviewer 1 Report

In the paper "Optimization of Phenolic Extraction from Helichrysum italicum  Using Cyclodextrins for Preparation of the Extracts with Anti-  Elastase and Anti-Collagenase Properties", the influence of extraction conditions on the content of phenolic compounds was well investigated and reported. Among the tested CDs, 2-hydroxylpropyl-beta-CD was  used to prepare two extracts with highest TPA, TP, and TF.  Anti-elastase and anti-collagenase activities of the prepared  extracts were well documented by the authors.

However some minor revision were herein suggested: 

·         Please add the abbreviation UAE when referring to ultrasound-assisted extraction and use it in the main text.

·         Please insert and use abbreviations properly throughout the main text (for instance: lactic acid, glycerol, cyclodextrin

·         Line 73 “Among such solvents, a prominent place is occupied by aqueous solutions of cyclodextrins (CDs).”The authors refer to an aqueous solution of cyclodextrins as one of the new 'green' and sustainable solvents. Please revise this sentence without considering the aqueous solution of cyclodextrins as an alternative solvent, but as already widely reported in literature, as an alternative extraction method: “Cyclodextrin-Assisted Extraction Method”. Please revise the entire manuscript accordingly adding appropriate citations (for instance: “Cyclodextrin-Assisted Extraction Method as a Green Alternative to Increase the Isoflavone Yield from Trifolium pratensis L. Extract, Pharmaceutics. 2021, 13(5): 620;  Cyclodextrin-assisted extraction of phenolic compounds: Current research and future prospects, Trends in Food Science & Technology, 79, 2018, 19-27, or others).

·  Table 1: Please modify the second column label “solvent” in “Extraction Method” and provide the extraction condition in Table 1 caption.

·         Line74-76

·         The authors refer to cyclodextrins as “molecules that have a ring structure with hydrophobic cavity and hydrophilic exterior wall, that enables them to form complexes with various organic compounds, including those of natural origin” please revise this sentence with a proper description of chemical structure od cyclodextrin adding appropriate citations.

·         Table 4: Please specify the extraction time in table notes.

·         The authors stated that the lactic acid LA could affect the pH of the solvent used for the extraction, but there is also the possibility that lactic acid forms a natural deep eutectic solvent (NADES) with one of the amino acids from the plant material increasing the solvent capacity of the used mixture.

·         Are the authors sure that at 80°C the acoustic cavitation is still optimal? Please, to prove that the effect of ultrasound is really crucial at 80°C add the data of TP, TPA, TF recovered after a conventional extraction performed under magnetically stirring at the same temperature (80°C) and time using the same extraction medium.

Reviewer 2 Report

The manuscript is well presented and organized. I have one concern only. It is shown that the extraction with CDs is viable alternative to solvent extraction. I would assume that the CD complexes (page2, line 75) the Authors refer to are inclusion complexes. However, the inclusion of such compounds is never demonstrated; I would urge the Authors to provide support for the inclusion of targets compounds into the CD cavity or, at least, to add a couple of sentences based on literature findings showing that these CDs do form inclusion complexes (as for example explicitly demonstrated in Ref. 13). 

Minor comments

Table 1                         It is not clear what n (CD) is. Does it mean number of mmoles?

Page 6, line 277          However, similar to the … This explanation is a bit speculative and should be softened.

Page 8, line 314          The third decimal place is given for all values expressed as mg/mL. The third decimal place is also given for SANA (page 5, line 215) while for other concentrations (see for example elastase, page 5, line 212) only the second digit is given. Shouldn’t all these values be consistently expressed?

There are a couple of English mistakes that I have taken the liberty to list below.

Page 2, line 61            by α-pinene. I guess this should be ‘in α-pinene’.

Page 3, line 103          I guess the left parenthesis should be placed just before Munich.

Page 6, line 229          I guess prepared would sound better.

Page 6, line 232          colagenase. This should be collagenase.

Page 6, line 271          are line. I imagine this should be ‘are in line’.

Page 8, line313          Why not simply ‘from Table 2’?

Page 8, line 349          deionization. Deprotonation would be a more appropriate term.

Page 14, line 478        as: phenolic. Is the colon necessary?

Reviewer 3 Report

The manuscript entitled "Optimization of Phenolic Extraction from Helichrysum italicum Using Cyclodextrins for Preparation of the Extracts with Anti-Elastase and Anti-Collagenase Properties" aimed to optimize the extraction of phenolic compounds from Helichrysum italicum with anti-elastase and anti-collagenase effects using cyclodextrins. The manuscript seems interesting and encompasses a hot research topic. However, some questions/suggestions should be addressed before considering its publication in Metabolites journal:

1. Abstract: Please indicate some results in the abstract.

2. Methods: Please indicate how was the extract dissolved (solvent and concentration prepared) for testing in spectrophotometric assays, namely for TP, TPA, TF, anti-elastase and anti-collagenase activities. 

3. Why were two models of response surface methodology (namely 2-Level Factorial Design and Box-Behnken Design) employed in the optimization of extraction conditions? Please explain and clarify it in the manuscript.

4. Methods: Why did not the authors quantify the phenolic compounds identified by LC-PDA-ESI-MS? Please explain.

5. Conclusion: In lines 603-605, what about the costs of the extraction technique employed? Is this extraction process economically viable for industrial applications? Please discuss it briefly. In addition, discuss further perspectives in the research field.

6. References: Please revise the references according to the journal guidelines. Scientific names of plants should be formatted in italic letter. Some journal names are in abbreviated form and others in full form. 

Minor comments:

- Line 25: "suited" or suitable?

- Line 26: Define the abbreviation "OPT".

- Line 27: Define the abbreviation "HP-β-CD".

- Line 31: Revise the sentence as "...positive controls, namely ursolic and gallic acids."

- Lines 31-32: Revise the sentence "This activity deems the prepared extracts as promising ingredients for natural cosmetics...".

- Line 38: Add "and" before "represents".

- Line 49: Format "in vivo" in italic letter.

- Line 50: Please consider deleting "abundance of".

- Lines 51-52: Consider placing "such as rosmarinic, neochlorogenic, isochlorogenic B, cichoric [2] 3,4-dicaffeoylquinic acid, chlorogenic acid, 3,5-dicaffeoylquinic acid [5]" within parentheses.

- Line 54: Also consider placing "such as rutoside [2], tiliroside [6] and kaempferol 3-O-glucopyranoside [5]" within parentheses.

- Lines 56-57: Please indicate some pharmacological activities.

- Lines 57-59: Define the abbreviations "NF-κB", "HIV", "IL", "TNF-α" and "PGE₂".

- Line 61: Delete "by" before "α-pinene".

- Line 80: Use "CDs" instead of "Cyclodextrins".

- Lines 91-95: Please highlight the novelty and relevance of this study to the research field.

- Line 117: Add a space between "2020" and "in".

- Please place Tables and Figures immediately after they are first mentioned in the manuscript. 

- Line 135: Please define the abbreviations "TP", "TF" and "TPA" where they first appear in the manuscript. 

- Line 146: Define the abbreviations "LA" and "GL" where they first appear in the manuscript.

- Lines 199: Rewrite "µL".

- Line 219: Add "the" before "respective". Also, why was ursolic acid used as standard for elastase inhibitory assay?

- Line 232: Correct "collagenase".

- Line 235: Why was gallic acid used as standard for collagenase inhibitory assay?

- Lines 236-237: Correct to "collagenase" (instead of "elastase").

- Lines 246-251: Add references to support these statements.

- Lines 252-254: Revise the sentence.

- Line 260: Correct "CD" to "CDs". Delete comma after "CDs solution".

- Line 261:Add "that" before "the extraction". Delete "that" before "matched".

- Lines 275-277: Please add at least one reference to support this statement.

- Line 278: Delete "of" before "herbal".

- Line 290: Delete "it is" before "UAE".

- Lines 340, 341 and 382: Replace "glycerol" by "GL".

- Figure 3: The authors should use the same letters presented in the text (X1, X2, X3, etc.) to identify the variables.

- Line 409: Delete "of" before "target phenolics".

- Line 414: Please consider indicating the extraction conditions for each run cited in the discussion within parentheses. Revise it along the manuscript.

- Line 447: Add "for TPA" ("...while R2 for TPA was somewhat lower...").

- Line 451: In the legend of Table 5, replace "content" by "contents" (plural) and delete "extractions" at the end of the legend.

- Line 468: Add "of" after "quantity".

- Line 499: Correct "kaempferol".

- Line 523: Define the abbreviation "UV".

- Line 528: Correct "anti-ageing" to "anti-aging".

- Lines 530, 533, 569 and 589: Format "in vitro" in italic letter.

- Lines 537: Format "in vivo" in italic letter.

- Line 593: Define the abbreviation "UVA".

- Line 599: Add "the" before "highest" ("with the highest").

Reviewer 4 Report

The paper titled: “Optimization of Phenolic Extraction from Helichrysum italicum Using Cyclodextrins for Preparation of the Extracts with Anti- Elastase and Anti-Collagenase Properties”, represent a valid and interesting work. The manuscript is well written and in my opinion can be considered for the publication in this Journal. Only some changes are required:

-        Pag. 2, line 117, H. italicum was collected in June 2020in… enter the space

-        Pag.3, line 129, I suggest you move Table 1 and insert it immediately after you mention it.

-        Pag. 3, line 150 and pag. 9, lines 333-337.  I advise the authors for a reason of readable to divide table 2, between the extraction conditions (and to enter it in the table under you cited it), and the results in other table (example table 3) with you shown the results.

-        I recommend that you also check all the other tables, because often in reading the text I find difficulties.

-        Review the instructions for authors.

In addition, I recommend you read these papers and if you think it useful you can possibly quote them.

De Bruno, A.; Romeo, R.; Fedele, F.L.; Sicari, A.; Piscopo, A.; Poiana, M. Antioxidant activity shown by olive pomace extracts. J. Environ. Sci. Health B 2018, 53, 526–533.

Imeneo, V.; Romeo, R.; De Bruno, A.; Piscopo, A. Green-sustainable extraction techniques for the recovery of antioxidant compounds from “citrus Limon” by-products. J. Environ. Sci. Health Part B 2022, 57, 220–232. https://doi.org/10.1080/03601234.2022.2046993.
